# Evaluation of Novel Doxorubicin-Loaded Magnetic Wax Nanocomposite Vehicles as Cancer Combinatorial Therapy Agents

**DOI:** 10.3390/pharmaceutics12070637

**Published:** 2020-07-07

**Authors:** Julia Jiménez-López, Lorena García-Hevia, Consolación Melguizo, Jose Prados, Manuel Bañobre-López, Juan Gallo

**Affiliations:** 1Institute of Biopathology and Regenerative Medicine (IBIM9090325ER), Center of Biomedical Research (CIBM), University of Granada, 18014 Granada, Spain; julia.jimlop@gmail.com (J.J.-L.); melguizo@ugr.es (C.M.); jcprados@ugr.es (J.P.); 2Department of Anatomy and Embryology, Faculty of Medicine, University of Granada, 18014 Granada, Spain; 3Biosanitary Institute of Granada (ibs.GRANADA), University of Granada, 18014 Granada, Spain; 4Advanced (Magnetic) Theranostic Nanostructures Lab, Health Cluster, International Iberian Nanotechnology Laboratory, Av. Mestre José Veiga s/n, 4715-330 Braga, Portugal; lgarcia@idival.org; 5The Scuola Superiore Sant’Anna, the BioRobotics Institute, Viale Rinaldo Piaggio 34, Pontedera, 56025 Pisa, Italy

**Keywords:** magnetic nanocomposites, magnetic hyperthermia, combinatorial therapy, 3D in vitro models, drug delivery

## Abstract

The development of nanotechnology-based solutions for cancer at a preclinical level advances at an astounding pace. So far, clinical translation of these new developments has not been able to keep the pace due to a range of different reasons. One of them is the mismatch between in vitro and in vivo results coming from the expected difference in complexity. To overcome this problem, extensive characterisation using advanced in vitro models can lead to stronger preliminary data to face in vivo tests. Here, a comprehensive in vitro validation of a combinatorial therapy nanoformulation against solid tumours is presented. The information extracted from the different in vitro models highlights the importance of advanced 3D models to fully understand the potential of this type of complex drugs.

## 1. Introduction

With the advent of nanotechnology, several fields have been posed to benefit from the altered physicochemical properties of matter at the nanoscale. One of the areas in which a deeper impact was predicted is human health [1]. The promise in this field, particularly in the area of oncology, already begun to be realised with products such as Doxil^®^, Myocet^®^ and Abraxane^®^, [2,3,4] examples of multimillion-euro oncological drugs based on nanotechnology formulations that were introduced in the clinic in the late 1990s and early 2000s. The advantages of these nanoformulations were clear, coming mainly from an increased bioavailability of the drug and reduced side effects. These early successes in the field encouraged researchers worldwide to pursue nanotechnology as a means to increase the efficiency of drugs. Despite the large number of publications yearly in the field of drug discovery and drug delivery (e.g., 89,736 papers, Web of Science “drug delivery” search last 5 years, 11/2019), the number of nanotechnology-based products that have reached the market is still limited [5,6]. A number of reasons can account for this apparent mismatch between research production and industry outputs, from economic factors to policy restrictions to patient underperformance issues [7]. Related to this last point, screening methodologies used for the initial biological evaluation of pharmacological formulations are usually restricted to the validation of the drugs in vitro with immortalised cell lines in classic 2D cultures. This methodology is widely implemented due to its suitable properties as first screening checkpoint (simplicity, reasonable cost, high throughput, possibility to follow different parameters/responses, etc.). However, it is evident that these simple 2D models are too far from representing the complexity of the real living organism. Other cellular models have been proposed as more advanced alternatives or complementary models to classic 2D systems [8,9,10]. Among them, multicellular tumour spheroids (MCTSs) can be used to mimic relevant characteristics of complex biological organisms such as, for example, the dense extracellular matrix, cellular heterogeneity, and cell–cell 3D interaction/communication [11,12]. In particular, in oncology, MTCSs can simulate more accurately the tumour microenvironment and the heterogeneity found within solid tumours in terms of O_2_ distribution and viability (necrotic core). Additionally, in oncology, the penetration of drugs/formulations into the tumour is a key parameter to study that is not accessible through 2D culture models. This parameter can be of particular interest with nanoformulations due to the “large” size (up to 500 nm) of nanostructures when compared to drug molecules. While, on one hand, this “large” size can be a drawback for the formulation to diffuse inside solid tumours, on the other, the well-known enhanced permeability and retention (EPR) effect will favour their accumulation in tumours [13]. Thus, a reliable model that takes the 3D solid structure of tumours into account is key for the swift development of chemotherapeutic nanoformulations. Likewise, formulations designed to deliver a combinatorial treatment will immediately benefit from more complex testing models where their full potential can be evaluated. This subject, the development of formulations where several therapeutic modalities are combined, is where nanotechnology becomes particularly interesting through the ability of delivery systems to encapsulate large amounts of drugs [14,15,16]. The high loading capacity of nanoformulations enables the simultaneous use of several drugs or drugs and effectors (e.g., photodynamic therapy, thermal therapy), sharing pharmacodynamic and pharmacokinetic profiles. The development of hybrid drug delivery systems incorporating an organic matrix (the encapsulating part) plus an inorganic component (usually the effector or reporter) is contributing greatly to the advancement of this field [17]. Of particular interest are magnetic hybrid systems in which imaging capabilities by MRI can be combined with drug delivery [18,19]. Furthermore, in these systems, magnetic hyperthermia can also be explored as a mean to control the release profile of a drug externally and influence (enhance) the therapeutic efficiency of the system, as we demonstrate in this work.

Here, we present the thorough in vitro validation of one of such hybrid delivery systems, a magnetic wax nanocomposite vehicle (mWNV), as a combined chemotherapy and thermotherapy formulation. These mWNVs are capable of delivering a traditional chemotherapeutic agent (doxorubicin, DOX) with an extra level of control over the release profile through external stimulation via magnetic hyperthermia. The combination of 2D and 3D cell culture models allows for a deeper understanding of the contributions, performance and potential of each therapeutic modality and their combination.

## 2. Materials and Methods

### 2.1. Preparation and Characterisation of Magnetic Wax Nanocomposite Vehicles (mWNVs)

The preparation and in depth characterisation of mWNVs is described in our previous publications [20]. Briefly, a modified melt emulsification method was used for the preparation of the mWNVs. In a normal preparation, 200 mg of carnauba wax were mixed in a glass vial with a chloroform solution of Fe_3_O_4_ nanoparticles (prepared following standard coprecipitation protocols [20]) containing 40 mg of Fe. To this solution, 250 µL of a chloroform solution of DiO (1 mg/mL) were added, followed by a chloroform solution of DOX (40 mg DOX, 1 mL). This mixture was heated under a heat gun until all the chloroform had evaporated and the wax melted. At this point, 4.5 mL of milliQ water followed by 0.5 mL of a water solution of Tween80 (50 mg/mL) were added to the vial and the sample was ultrasonicated for 2 min at 25% power at 20 s working intervals. Immediately after the sonication, the vial was immersed in ice to solidify the lipid nanoparticles. Once cold, the formulation was centrifuged (956× g, 10 min), the pellet discarded, and the supernatant freeze dried in the presence of sucrose (0.9% w/w) as cryoprotectant. The mWNVs were characterised via a combination of techniques including DLS, TEM, XRD, UV-Vis, FTIR and fluorescence spectroscopies, ICP-OES, SQUID, HPLC, relaxometry, MRI and calorimetry (Appendix A).

### 2.2. Cell Culture

Human breast cancer (Hs578T), murine melanoma (B16F10), resistant HCT-15 (high P-gp expression) and sensitive T84 (low P-gp expression) colon cancer cells were grown in Dulbecco’s Modified Eagle’s Medium (DMEM) (Sigma-Aldrich, St Louis, MO, USA) supplemented with 10% fetal bovine serum (FBS) and 1% of penicillin-streptomycin (Sigma-Aldrich). Cells were maintained at 37 °C in a humidified incubator with a 5% CO_2_ atmosphere.

### 2.3. Proliferation Assay

Cell proliferation was determined using a sulforhodamine B (SRB) assay [21]. 5 × 10^3^ Hs578T cells were plated in 24-well plates with respective culture medium and incubated overnight. Then, cells were treated with DOX and DOX loaded NPs as well as the equivalent volume of NPs without DOX at concentrations ranging from 0.5 to 10 μg/mL for 24 and 48 h. After the incubation, cells were fixed with 10% trichloroacetic acid (TCA), washed three times with distilled water, dried overnight and stained with 0.4% SRB in 1% acetic acid solution. The dye was resuspended with a 10 mM pH 10.5 solution of Trizma and quantified at 492 nm in a Biotek Synergy H1 plate reader. The percent cell viability was calculated considering the untreated cells as 100% viability.
(1)$$Cell\;viability\;\left(\%\right)=\frac{Treated\;cells\;OD-blank}{Control\;OD-blank}\times 100$$

The results were used to determine the half maximal inhibitory concentration *IC*_50_ and the *Therapeutic index (TI)* using the next equation:(2)$$Therapeutic\;index\;\left(TI\right)=\frac{DOX\;IC_{50}}{mWNV-DOX\;IC_{50}}$$

### 2.4. Flow Cytometry Analysis

The effects of mWNVs, mWNV-DOX and DOX in Hs578T and B16F10 cells were analysed by flow cytometry. A total of 10,000 fixed and stained cells per condition were used. Flow cytometry analysis was carried out in a CytoFLEX equipment (Bectam Coulter, Brea, CA, USA), where a total of 10^4^ events per sample were acquired. Flow cytometry data were processed and analysed using the CytExpert Software (Beckman Coulter Brea, CA, USA).

### 2.5. Multidrug Resistance Assay

The ability of the mWNV formulation to overcome multidrug resistance (MDR) based on P-glycoprotein was evaluated in resistant (HCT-15) and sensitive (T84) colon cancer cell lines using a SRB assay. Firstly, cells were seed in 24-well plates at a density of 6 × 10^4^ and 4.5 × 10^4,^ respectively, and treated with free DOX and mWNV-DOX, as well as the equivalent volume of unloaded mWNVs. In parallel, cells were pretreated with the P-gp inhibitor verapamil (Sigma-Aldrich) (14.3 µM) 24 h before the administration of the treatments. After 48 h, cells were fixed with 4% TCA and stained with SBR. Cell viability and IC_50_ were calculated as described above. 

### 2.6. Confocal Microscopy

Cells and 3D-MCTS were fixed in 4% paraformaldehyde, actin was stained with phalloidin-tetramethylrhodamine B isothiocyanate (Sigma-Aldrich) and DNA with DAPI. Confocal microscopy images were obtained with a Carl Zeiss inverted microscope attached to the LSM 780 confocal system (software; ZEN 2010).

### 2.7. D-MCTS Growth Inhibition Assays

A 96-well plate was coated with 1% of agarose (100 µL per well) and left to dry for 30 min. Then, cells were harvested by trypsin and 1 × 10^4^ and 1.5 × 10^4^ Hs578T and B16F10 cells were seeded onto this 96-well plate respectively in a final volume of 150 µL of media per well. The plate was centrifuged at 800 g during 30 min in order to promote cell aggregation into the well. After 72 h of incubation, MCTS were treated with DOX, mWNV-DOX and unloaded mWNVs at the dose of 10 and 2 µg/mL in Hs578T and B16F10 cells, respectively. All tests were done in octuplicate. In parallel, MCTS treated with the different treatments were exposed to magnetic hyperthermia (240 G, 554 KHz) for one hour. MCTS were then grown for 7 days at 37 °C in a 5% CO_2_ atmosphere incubator and 50% of the media was exchanged every two to three days. MCTS growth was monitored every 2 days using an inverted phase-contrast microscope (Nikon Eclipse TS100, Nikon Corporation, Tokyo, Japan) connected to a digital camera (Nikon, DS-Fi1) and their longest (*LM*) and shortest (*Lm*) length were measured with ImageJ software to obtain a median relative volume (*V*, μm^3^) using the following equation [22]:(3)$$V=\frac{LM\times Lm^{2}\times \pi }{6}$$

### 2.8. Statistical Analysis

Student’s *t*-test was used for the statistical analysis; differences were considered significant with a *p*-value < 0.05 and lower.

## 3. Results

### 3.1. Preparation and Characterization of DOX-Loaded Magnetic Wax Nanocomposite Vehicles (mWNV-DOX)

The nanocomposites object of this study (Scheme 1) were prepared in one step following a modified melt emulsification method according to our published procedure [20,23]. Basically, an organic phase composed of the wax, drug (DOX), magnetic nanoparticles and fluorescent dye (DiO) was sonicated at temperatures above the melting point of the wax in the presence of a water phase composed of water and a surfactant (Tween80). For the preparation of this nanocomposite, a safe-by-design approach was followed; all the major components (wax, surfactant, drug and iron oxide-based magnetic nanoparticles) are FDA-approved for human use. The melt emulsification protocol allowed us to obtain mWNVs of around 200 nm of hydrodynamic diameter with promising theranostic properties (Scheme 1) [20].

The encapsulation of DOX into the mWNVs does not have a dramatic effect on most of the physicochemical parameters of the formulation; the main difference occurs at the level of the zeta potential of the particles that undergoes inversion from fairly negative values (−56 mV) to positive values (+12 mV) due to the positive charge of DOX at physiological pH (pKa 8.4, PubChem). The properties of the formulation as MRI contrast agent and as MH effector remain mostly unchanged and present values superior to those of commercial contrast agents [24]. The high stability of these mWNVs is highlighted by the possibility of direct TEM imaging without the use of staining with heavy metals or cryo-techniques (Appendix A).

### 3.2. Vehicle Biocompatibility in 2D Cultured Cells

The main aim of this study was to explore, through a thorough in vitro validation in classic 2D and advanced 3D models, the therapeutic potential of the proposed mWNV-DOX formulation in cancer. For this purpose, two different tumour cell lines were selected as starting points. On one hand, a human triple negative breast cancer cell line, Hs578T, was selected as a well-established example of malignancy currently treated with DOX. On the other, B16F10, a mouse melanoma cell line, was selected as an example of malignancy that could benefit from an improved therapeutic strategy based on DOX (particularly in the case of metastasis).

First, the potential toxicity of the vehicle (mWNV) was evaluated by incubating both types of cells with serial dilutions of the control formulation for 48 h. The range of concentrations covered in these tests was significantly larger than the concentrations used in following tests involving the drug. As shown in Figure 1a, no toxicity was detected, even at 100 µg/mL of the formulation on either cell line. These results were also validated by flow cytometry (Appendix A); no significant impact was observed in the cell cycle status of the cells. Further, confocal imaging studies confirmed both the internalisation of mWNVs into both cell types (Figure 1b) and their lack of impact on cell phenotype. mWNVs are clearly visible as individual green spots (from DiO in the formulation) in the proximity of the nucleus of the cells. Together, these results guarantee the biocompatibility of the vehicles at a cellular level.

### 3.3. Chemotherapeutic Effect in 2D Culture Cells

Once the biocompatibility of the formulation at the cellular level was verified, cell viability tests were performed with DOX-loaded mWNVs using serial dilutions of the formulation at two different time points, 24 and 48 h. The results obtained were compared to the effect of free DOX to evaluate the influence of the encapsulation. As expected, the effect of DOX (free and encapsulated) was different on each cell line (Figure 2 and Table 1), being that melanoma cells are, in general, more sensitive to DOX, whether encapsulated or not. In all cases, the encapsulation of DOX in mWNVs brought a positive effect in the form of a significant decrease of IC_50_ values. This effect is more evident at shorter time points; the encapsulation reduced the IC_50_ over 80% for Hs578T cells and around 40% for B16F10 after 24 h, which translates into a therapeutic index (TI) of 7.53 and 1.75, respectively. However, at 48 h, differences in IC_50_ between DOX and mWNV-DOX are reduced (TI of 3.62 in Hs578T and 1.33 for B16F10). This time dependency points towards a more efficient/faster uptake and delivery of DOX to the nucleus of the cells by the mWNVs. Confocal microscopy studies following DOX inherent fluorescence were performed to support this hypothesis. Data shown in Appendix A clearly confirm this theory. During the first 6 h of incubation, the accumulation of DOX in the nucleus is 38% higher in the case of mWNV-DOX than of DOX. As before, these results were also validated by flow cytometry (Appendix A); a clear dose-dependent impact was observed in the cell cycle status of the cells in accordance to the data shown in Figure 2.

### 3.4. Encapsulation Effect on a DOX-Resistant In Vitro Model

The enhanced cytotoxicity of the mWNV-DOX, related to a faster uptake, encouraged us to test the mWNV-DOX in a model of chemotherapy resistance.

Several mechanisms have been described as potentially responsible for chemotherapy resistance, including drug inactivation, target alteration, enhanced DNA repair, cell death inhibition, decreased drug uptake and drug efflux, among others [25,26]. In fact, one of the better-studied factors responsible for multidrug resistance is a membrane transporter protein called P-glycoprotein (P-gp) that is responsible for pumping a range of drugs out of the cell. Verapamil is a calcium channel blocker that binds to P-gp competitively with respect to antineoplastic drugs, inhibiting the excretion of anticancer drugs (Figure 3a). In the last several years, novel therapeutic approaches using drug-loaded nanoparticles and verapamil have been developed to increase intracellular drug concentration while minimizing toxicity [25,26,27].

In this context, a cellular model in which drug efflux through P-glycoprotein is characterised as being a major responsible for resistance (HCT-15) [28] was selected, together with a sensitive cell line of the same typology (T84) as the control. Cell viability in both cell lines was tested after their exposure to increasing concentrations of DOX, mWNV-DOX and their combination with verapamil (Figure 3b–e). While in T84-sensitive cells, the difference in viability between DOX and encapsulated DOX is only significant at high doses (>5 µg/mL, Figure 3b), there is a significant difference in all concentrations tested on resistant cells (HCT-15), showing an enhancement of antitumour activity caused by the encapsulation (Figure 3d). In fact, the IC_50_ is reduced in more than 80% with the encapsulation in the resistant cell line versus only 45% in the sensitive one (Table 1). Specifically, in the resistant line the encapsulation of the DOX produces a significant and pronounced reduction in the IC_50_ of the drug, from 56.4 µg/mL to 9.5 µg/mL. As predicted, when verapamil was used to inhibit P-gp, the IC_50_ values of mWNVs-DOX, and particularly of free DOX, were strongly reduced (from 56.4 to 1.8 µg/mL for DOX, and from 9.5 to 2.0 µg/mL in mWNV-DOX).

### 3.5. Combinatorial DOX-MH Treatment in 2D Cultured Cells

The presence of magnetic nanoparticles in the mWNVs responds to different needs as mentioned above. One of their intended applications is their use as magnetic hyperthermia effectors. The combination of DOX therapy with MH controlled release (Figure 4) showed positive synergic effects on both cell lines. In B16F10 cells, cell viability dropped from 70% to 25% upon the application of MH for 1h (0.5 µg DOX/mL) (Figure 4a,c), while in Hs578T cells, the viability dropped from 55% to 20% under the same conditions (Figure 4b,d). In addition, a cell toxicity of virtually 100% could be achieved at DOX concentrations of 1 µg/mL with the application of 1h MH for both cell lines. While the role of MH in combination with DOX delivery is clearly demonstrated by these tests, the effect of MH as treatment on its own (using unloaded mWNVs) was negligible on the cells (Appendix A), further highlighting the potential of this combinatorial treatment scheme.

3.6. mWNV-DOX in 3D MTCSs

The above results show the effects of mWNV-DOX nanocomposites on tumour cells in classic 2D models. This is a very simple and straightforward technique to evaluate the effect of a compound on the viability of different cells, but it does not represent accurately enough the potential of a given drug/combination in vivo. A higher level of complexity is required to better understand the therapeutic potential of a given treatment. Multicellular tumour spheroids are still a simple model to screen and study cytotoxic drugs, but they already incorporate some of the features lacking in 2D models. Thus, mWNVs were next tested in MCTS of B16F10 and Hs578T cells. Hs578T and B16F10 cells were grown over 1% agarose gels to develop 3D MCTS over two to three days and then they were treated once with DOX, control mWNVs and mWNV-DOX, and their size was monitored for seven days. Their volume was followed every other day during the treatment and the penetration and DOX release/accumulation was studied via confocal imaging (Figure 5 and Appendix A).

First, free DOX presented a positive effect on the growth inhibition of both cell lines, although this effect was more evident in B16F10 cells than in Hs578T cells. Specifically, B16F10 MTCSs showed significant reductions in size from day 3 with respect to control untreated MTCSs (Figure 5a), while in Hs578T, this significant reduction was only observed at day 7 (Figure 5b). The incubation with control unloaded mWNVs did not present any deleterious effect on either of the MCTSs, as in the case of 2D tests (Appendix A). However, 2D tests showed an improved performance of mWNV-DOX versus DOX that was lost in 3D MCTSs. The effect of DOX, although different for each cell line, was statistically the same whether encapsulated or free (70% volume reduction in B16F10 and 16% in Hs578T cells with respect to control MCTSs, Figure 5a,b). To bring some light into this lack of differences, the uptake and localisation of DOX was investigated via LSM. As shown in Figure 5c, mWNV-DOX (green) were able to penetrate the MCTSs and release DOX (red) into cells in a more homogeneous manner than free DOX. Confocal images of MCTSs treated with DOX show DOX present in localized peripheral areas of the MCTSs. A simple analysis of LSM images on DOX distribution within the MCTSs clearly shows a more homogeneous distribution when the drug is applied encapsulated as compared to the free drug (Appendix A).

Next, the combination of DOX treatment with MH controlled released was tested on MTCSs. As seen in Figure 6 and Appendix A, the application of alternating magnetic fields combined with DOX encapsulation has a beneficial effect on treatment outcome (measured as MTCS volume). While no direct effect of MH on its own was observed in 2D cultures, in 3D MTCSs, a reduction in size is observed from day 3 in both cell lines, even in the absence of DOX (Figure 6a,e). It seems, however, that at longer time points, the inhibitory effect of a single MH treatment on cell growth is not enough to prevent MTCS relapse (day 7 in B16F10). The combination of MH with encapsulated DOX delivery had a more powerful effect, being able to revert and inhibit MTCSs growth consistently from day 1 and 3 for B16F10 and Hs578T cells, respectively, until the end of the tests. Controls performed to test the potential effect of the MH treatment itself were negative as expected, meaning that application of MH to cells treated only with free DOX does not change the efficiency of the drug (Appendix A). Compared to the effect of DOX, the combination of mWNV-DOX and MH presents a synergic effect on MTCSs growth from day 1 (Figure 6c,g, Appendix A), bringing an additional reduction of size very similar in both systems of around 35% (from 30% control to 19% mWNV-DOX + MH in B16F10 and from 84% control to 58% mWNV-DOX + MH in Hs578T, Table 2). It is interesting to notice that even though the overall effect was very different between cell lines (stronger in B16F10 in which the reduction in volume led to MTCS 70% smaller than control versus only 16% reduction in Hs578T), the improvement due to the combination of drug delivery with MH was very similar in both systems (37% in B16F10 versus 31% in Hs578T). It is also interesting to notice that while conventional 2D systems reported a higher improvement of the therapy with encapsulation for Hs578T cells (72% vs 25% for B16F10), 3D systems showed nearly no differences between both cell lines. More complex 3D systems allowed to observe differences from this complex system, even differences due to the combination of MH with mWNVs not only when DOX was present, but also in controls without the drug.

## 4. Discussion

The classic application of magnetic hyperthermia uses heat generation from magnetic nanoparticles as a direct means to kill cancer cells (thermal ablation). A straightforward limitation of this strategy is the need for local injection of the nanoparticles to reach a threshold concentration sufficient for the complete thermal ablation of the diseased tissue. In our case, thermal ablation is only used as a collateral effect to the triggered thermal release of DOX into the tumour. MNPs act as external stimuli-responsive switches to add an extra level of control and safety over the DOX release profile from the mWNVs [20]. Under the MH conditions used for this study (see M&M), the release of DOX can be locally increased on demand through the application of alternating magnetic fields to impact diseased tissues while preserving healthy ones. The drug of choice in this work, DOX, was selected not only because it is FDA-approved and commonly used in the clinical practice [23] as mentioned above, but also for its favourable physicochemical properties (ideal hydrophobicity, red fluorescent nature) and current limitations in its applicability, mainly severe side effects (e.g., cardiotoxicity) [24] that can potentially be overcome through smart delivery approaches.

Regarding the preparation of the mWNVs, the protocol followed here produced particles around 200 nm in hydrodynamic diameter, which is adequate for systemic administration. The relatively large ζpot ensures the colloidal stability of the particles, and the functional performance of the mWNVs in both MRI and MH was superior to that of commercial agents. The relaxivity (*r*_2_) value of the mWNVs (up to 5 times greater than that of commercial *T*_2_ contrast agents) [24] enables their use as MRI contrast agents to follow the accumulation of the formulation in the tumours and the progression of the treatment. Likewise, the high SAR (specific absorption rate) measured for the mWNVs (see table Scheme 1) guaranties the strong production of heat from the magnetic nanoparticles (SAR value 4 times higher than Feridex) [29].

Traditional in vitro biocompatibility tests showed that control mWNVs without DOX were perfectly tolerated by cells to concentrations larger than the ones needed to obtain a therapeutic effect. These tests also demonstrated that cells are capable of internalising these vehicles efficiently and that, phenotypically, they do not induce substantial changes on the cells. However, when DOX is incorporated into the mWNVs, the therapeutic index of such formulations increases compared to that of free DOX. This enhanced therapeutic effect was found to come from the efficient mWNVs uptake by the cells and subsequent drug release, as revealed by confocal imaging of DOX accumulation in the nucleus of the cells. Even though DOX can cross the cell membrane and accumulate in the nucleus at a sufficient speed to have an impact on cell survival, its incorporation into mWNVs can speed up this process due to the good internalisation of mWNVs. This effect prompted us to test mWNVs-DOX in a model of drug resistance controlled by P-gp (an ATP-dependent transporter reported to confer resistance to a variety of drugs such as vinblastine, paclitaxel and DOX, [27] and being expressed in several types of cancer [28]), as per each internalisation event of an mWNV, a number of DOX molecules in the order of 10^5^ are internalised, which can potentially overload P-gp and thus overcome resistance. This model further supported the advantages of mWNVs as drug delivery vehicles, as their positive effect was confirmed and supported by control experiments using P-gp inhibitor verapamil (one of the most extensively studied, having even entered clinical trials [29]). Therefore, we wanted to examine the effect of mWNV-DOX chemotherapy in a resistant cell line. In the absence of the P-gp inhibitor, mWNV-DOX were able to reduce the viability of DOX-resistant cells over 80% compared to free DOX. Interestingly, the effect of verapamil on cell viability was so pronounced that differences in performance between DOX and mWNV-DOX were lost (therapeutic index of 0.9), highlighting the role of P-gp on chemotherapeutic resistance in this model. On the other hand, the application of verapamil on the DOX-sensitive cell line (T84) did not bring any dramatic differences in terms of IC_50_ to DOX nor mWNV-DOX. These combined results show a positive effect of DOX encapsulation on the therapeutic outcome in vitro in cases of P-gp mediated resistance, pointing possibly to a drug overload of the transporter when DOX is delivered encapsulated and released intracellularly.

The combination of mWNVs-DOX with magnetic hyperthermia stimulation was then tested in traditional 2D cultures. A clear synergistic effect was observed in all cell lines and concentrations tested were the combination of 1 μg/mL of encapsulated DOX with 1 h MH stimulation led to virtually 100% cell death in both models. These promising results suggest that mWNVs-DOX would be good candidates in combination with MH as antitumour therapeutics. However, we believe that the acquisition of further data using more advanced model systems will provide more information and benefit decision-making on whether these systems are suitable for next stage of in vivo preclinical testing. Here is where multicellular tumour spheroids can make a significant contribution. MCTSs were prepared of both cell lines and first the effect of mWNVs-DOX was tested without the application of MH. Under this conditions, both mWNVs-DOX and free DOX showed a positive effect on MTCSs growth inhibition (more pronounced in B16F10 cells than in Hs578T cells), but, unlike the case of the 2D models in which the advantage of encapsulation was clear, in these 3D models, there was no significant difference in growth inhibition between both groups. To try to explain this lack of differences, confocal microscopy was used to study the distribution of DOX within the MTCSs. The distribution pattern of DOX from mWNVs-DOX was more homogeneous than that from free DOX. While mWNV-DOX can potentially reduce the size homogeneously in all regions of MCTSs, DOX seems to have a more local effect. This different ‘mechanism of action’ leads to no differences in terms of MCTSs overall volume reduction. More complex analysis will be required to elucidate the effect of the different distribution patterns of DOX in MCTSs, as simple size measurements cannot account for the differences observed in DOX penetration/distribution.

Finally, the effect of the combined chemotherapy/thermotherapy treatment was evaluated in MTCSs. In this case, compared to traditional 2D setups, 3D MTCSs allowed us to observe more detailed effects on cell growth/inhibition. In the case of MTCSs, the effect of MH on control mWNVs without DOX was significant compared to the control samples. The direct thermal ablation effect that was not observed in 2D cultures was measurable in 3D models. Furthermore, when the administration of mWNVs-DOX was combined with MH, there was a synergic effect on MCTSs growth inhibition superior to the effect of free DOX on its own.

## 5. Conclusions

Rapid screening of potential chemotherapeutic compounds and new therapeutic combinations using classic 2D cell cultures is a powerful technique used all over the word. However, the extrapolation of results obtained in vitro with 2D cell culture to in vivo is unrealistic due to the strong limitations of classic cultures. 3D cell culture and, particularly, multicellular tumour spheroids appear as a simple alternative, or better, as a complement to traditional cell culture. They bring into play a number of important features to better mimic the in vivo reality (cell communication, extracellular matrix, inhomogeneity). In this work, we have explored the potential of DOX-loaded magnetic wax nanocomposite vehicles as combinatorial therapy against two different types of malignant cells, and we have compared their performance in 2D and 3D MTCS models. It is interesting to see that while in 2D, the positive effect of encapsulation was clear, but in 3D, no differences could be measured. A different penetration pattern was observed for encapsulated vs. free DOX, but this did not lead to differences in volume. The combination of DOX delivery with magnetic hyperthermia provided an extra level of control over the drug release profile which in 2D systems translated into an enhanced cytotoxic effect on tumour cells. This effect was still observed in 3D MTCSs, with which a synergic effect was observed. It is also worth noticing that while in 2D models, a stronger performance was observed for Hs578T cells, but in 3D MTCSs, the opposite was true—the effect was more pronounced in B16F10 cells. The potential of the combinatorial therapeutic scheme was better described by 3D systems, where the effect of the different components of the system was better observed. Further studies are required to better understand and interpret data obtained from 2D and 3D cell models to be able to predict/infer the behaviour of therapeutic compounds in vivo.

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
