# Peer review of "Evaluation of Novel Doxorubicin-Loaded Magnetic Wax Nanocomposite Vehicles as Cancer Combinatorial Therapy Agents"

_pharmaceutics, 2020, doi:10.3390/pharmaceutics12070637_

Round 1
Reviewer 1 Report
The manuscript "Evaluation of novel magnetic wax nanocomposites vehicles as cancer combinatorial therapy agents" describes non only a novel nanoformulation but a thoughtful approach to evaluate the combination therapy in vitro, taking into account drawbacks for in vivo translation. The manuscript is well written and the results are could be interesting to the nanomedicine research community. Below are few comments to improve the manuscript:
- The authors mention the change in the zeta potential of the nanocomposite as indicative of DOX encapsulation. Can the authors elaborate more on this conclusion?. It is not very clear how the zeta potential will change after DOX encapsulation if in theory empty and DOX-encapsulated nanoparticles have the same surface composition.
- The phrase "...The high stability of these mWNVs is highlighted by the possibility of direct TEM imaging without the use of staining with heavy metals or cryo techniques.", is misleading to TEM results which are not shown. TEM or SEM image of the nanocomposite will be very helpful and perhaps necessary.
- Could the authors provide PDI for hydrodynamic diameters measured to confirm such "high stability"
- Statistical analysis for cell viability and chemotherapeutic effect in 2D culture cells needs to be performed.
- Doxorubicin abbreviation needs to be homogenized. Although most of the time is referred as DOX, in some parts of the text and Figures is abbreviated as Dox or Doxo.
Reviewer 2 Report
The manuscript titled “Evaluation of novel magnetic wax nanocomposite vehicles as cancer combinatorial therapy agents” has been reviewed where the authors have designed and presented an in vitro validation of a combinatorial therapy nanoformulation against solid tumours. However, there are a few concerns which are to be resolved prior to the acceptance of this work for publication.
- The authors have claimed that the vehicle is designed as nanocomposite, a concrete evidence is required for the claim. For this, TEM (Trans Electron Microscopy) or SEM (Scanning Electron Microscopy) images of Fe3O4 nanoparticles and magnetic wax nanocomposite would be convincing to show their size, structure and homogeneity.
- In line no. 85-86, temperature and duration of heating to be mentioned so that the data can be reproduced by others.
- Line No. 90. Mention ‘g’ valued instead of rpm as this will vary from instrument to instrument.
- In Line No. 92, authors have mentioned that they have characterized the materials using various techniques, e.g., TEM, XRD, FTIR, but they have not provided these data. These data are required in the manuscript as they have synthesized the nano materials.
- In the table shown in Scheme 1, the error values (±) are to be mentioned or include polydispersity index (PDI). Why the hydrodynamic value of mWNVs-DOX is smaller than that of mWNVs?
- In the caption of Scheme 1, ‘rt’ needs to be defined so that the caption becomes self-explanatory.
- In Figure 1, potential toxicity of vehicle (mWNV) was evaluated with an additional evidence of flow cytometric analysis of cell cycle status. Where as in Figure 2, cell viability was evaluated with DOX-loaded mWNV which shows reduced cell viability. Authors can provide flow cytometry analysis with DOX- loaded mWNV.
- A cell colony survival assay / colony formation assay can be provided to strengthen the drug effect with and without encapsulated DOX.
- In Figure 1 b, confocal microscopy images of both cell lines without mWNV (green dots) needs to be provided as control experiments.
- In Figure 2, free DOX has been used to evaluate cell viability. Since DOX is a chloroform solution, impact of chloroform has to be investigated too.
- If possible a simple western bolt with different cell cycles markers can be provided to support flow cytometric analysis of cell cycle status.
- Though authors used 1% soft agar assay to developed 3d-MCTs but a low attachment plate of sphere formation assay can be performed to evaluated sphere size, number, and volume to mimic 3D culture followed by combinational drug treatment which would strengthen their points.
Reviewer 3 Report
This manuscript "Evaluation of novel magnetic wax nanocomposite vehicles (mWNV) as cancer combinational therapy agents" claimed several advantages of mMNV carrying doxorubicin (D0X): improving delivery of DOX compared to free DOX and Magnetic hyperthermia effect.
1. First, modify title of this draft: vehicle itself carrying Fe3O4 cannot serve as "combination" therapy "agent".
2. It is strongly suggested that authors include the benefit of magnetic vehicles and current development/report of Magnetic hyperthermia in conjunction with chemotherapy. The current version include significant details of 2D and 3D cell culture. This has been widely reported elsewhere and it shouldn't be the main focus of this article.
3. Regarding table 1, please make sure you use "," or "." correctly.
4. How was the encapsulation improvement calculated? Is that a correct terminology? Encapsulation improvement gives me an impression of improvement in terms of formulations achieving greater solubility by incorporating drugs in the vehicle. Please justify this terminology and elucidate the equation.
5. Did authors test hyperthermia effect on control cells without treatment? This is an important control in data where magnetic hyperthermia effect is emphasized.
6. Figure 5 and 6 are confusing. Are some data points reappearing in other graphs?
For example, Dox from (A) first row on the right reappears on the second row. Same for mWNVs-DOX, which appears in three separate graphs.
Please combine data sets and present on one or two graphs.
7. Does shrinking the volume of 3D cell aggregates mean being treated? Have the authors actually counted the number of cells per spheroid post treatment? There may be a situation where cells aggregate even tighter after being treated whereas non-treated 3D spheroids spread somewhat widely. Increasing 3D spheroid volume by making it porous (with wider gaps) rather means "better" treated. Please comment on this matter with references if available.
8. What is the polydispersity index of nanoparticles?
9. Is there any innate toxicity of DOX-incapsulated mWNVs on cells due to its zeta potential, considering that mWNV itself is negatively charged?
10. Please include the method of Fe3O4 nanoparticles. Showing "prepared following standard co-precipitation protocol" without its own citation should not be appropriate.
11. Conclusion must be concise. Please remove data and edit this section appropriately.
12. Please remove unscientific expressions such as "..."
Reviewer 4 Report
It was a great pleasure to read this manuscript “Evaluation of novel magnetic wax nanocomposite vehicles as cancer combinatorial therapy agents”. The authors investigated the release of doxorubicine by cell viability assays for different release techniques with the focus on magnetic hyperthermia.
In the case of magnetic hyperthermia, I am very curious about the temperature, which is reached during the release in these experiments. Since magnetic hyperthermia itself can also contribute to cell death, the temperature should also be considered. Can you add the temperature measurements during the release? Is there any other release mechanism which helps to distribute the Dox aside from the melting of the wax?
Furthermore, the authors elaborate about the FDA approval of DOX. Are the other components used here also FDA approved? I do not say they need to be approved, but I am very interested in which materials are actually allowed for in vivo applications.
The introduction is nicely written and immediately triggered my curiosity. The citations are balanced and the results are well discussed.
A side from my previous questions, which should be addressed, I found some small erors:
Page2/3
The mWNVs were characterised via a combination of techniques including DLS, TEM, XRD, UV-Vis, FTIR and fluorescence spectroscopies, ICP-OES, SQUID, HPLC, relaxometry, MRI and calorimetry. You should cite here, where the wax has been characterized, even though you published the results.
Figure 6 y-axis labelling says volume instead of volume
In the supporting information:
Figure S1: Flow cytomtrey
Round 2
Reviewer 2 Report
In Line No. 95, it could be "956 x g" instead of 956 g as it might confuse readers as 956 gram and that is the correct way to represent centrifugation.
The manuscript might be accepted for publication with this minor correction.
Reviewer 3 Report
The authors made notable efforts to make improvement.
Here are a few minor comments:
Please spell out doxorubicin within the new title.
Please remove "..." in conclusions.
I still do believe that figure 5 a and b must be reorganized so that the same data sets are not repetitively displayed.
I do understand the author's point that some of the graphs may overlap on the same graph panel. This is also important to show how close each pattern is. The authors may use various colors, dotted lines, and solid lines to make it clearer.
